# Market Choices Driven by Reference Groups: A Comparison of Analytical and Simulation Results on Random Networks

**DOI:** 10.3390/e23081007

**Published:** 2021-08-01

**Authors:** Michał Ramsza

**Affiliations:** Department of Mathematics and Mathematical Economics, SGH Warsaw School of Economics, Al. Niepodległości 162, 02-554 Warsaw, Poland; mramsz@sgh.waw.pl; Tel.: +48-22-564-9257

**Keywords:** reference group influence, random graphs, simulations

## Abstract

The present paper reports simulation results for a simple model of reference group influence on market choices, e.g., brand selection. The model was simulated on three types of random graphs, Erdos–Renyi, Barabasi–Albert, and Watts–Strogatz. The estimates of equilibria based on the simulation results were compared to the equilibria of the theoretical model. It was verified that the simulations exhibited the same qualitative behavior as the theoretical model, and for graphs with high connectivity and low clustering, the quantitative predictions offered a viable approximation. These results allowed extending the results from the simple theoretical model to networks. Thus, by increasing the positive response towards the reference group, the third party may create a bistable situation with two equilibria at which respective brands dominate the market. This task is easier for large reference groups.

## 1. Introduction

The concept of the reference group is well known in social psychology, cf. [1,2,3]. A very wide definition was given in [4]. According to this definition, a reference group is an individual or a group, be it actual or imaginary, that has significant relevance to an individual’s evaluations, aspirations, or behavior. In other words, a reference group has a social influence, be it intentional or unintentional, on an individual’s behavior, cf. [5].

In social psychology, a social group is a group of individuals interacting with each other in various ways and capacities. A reference group is a specific frame of comparison to an individual. To model such a concept, one needs to model interactions between individuals. Graph theory has been successfully used to accomplish this task. An enormous body of literature covers the applications of graph theory to modeling interactions between various entities. In particular, graphs have been used to model social networks, information networks, technological networks, and biological networks, cf. [6] (Table 3.1, p. 182).

The research on graph applications to network modeling can be divided into several broad areas. Much ongoing research is concerned with the structure of networks and implied characteristics [6,7,8,9,10,11], dynamic models of network formation [12,13,14,15,16,17,18], and dynamic processes over static networks. The model presented in the current paper belongs to the latter type. The typical problems studied within this research area are social learning [18,19,20,21], opinion formation [14,22,23,24,25], information propagation [6,13,26,27,28,29], and the market consequences of these processes [4,30,31,32,33,34].

In [35], a simple model of the population making market choices was proposed. The proposed model fits into a strand of models present in the literature and mentioned above. In the model, a finite population of consumers makes repeated decisions about buying one of two brands. Each consumer is associated with a node in a graph of relations between consumers. The decision process of each consumer is based on his or her reference group given as a group of his or her direct neighbors.

The model can be thought of as the model of opinion formation, on the one hand, and information propagation, on the other. However, the model is different from the models present in the literature in two aspects. In the models of opinion formation, it is common to endogenize how the individual’s opinion depends on the views of others. In the model presented in [35], this dependence was based on the exogenous probability that potentially can be manipulated by the third party. This approach is dictated by some results related to influencing individuals’ attitudes through careful filtering of information [36]. The model also differs from the models of information propagation. These models are usually based on some diffusion processes, while some research is also concerned with finding the most influential individuals in a network. A single individual sends information to other individuals in such models, hence searching for the most influential individual. In the presented model, a group influences a single individual, and so, the mechanism is reversed from one-to-many to many-to-one.

The model presented briefly above leads to complex interdependencies. A group may influence a single individual, but this individual is a member of the reference groups of other individuals. How exactly such a structure is built depends on a graph’s topology. Thus, it is not possible to predict the behavior of such a model through a simple sum of individuals’ behaviors. Because of this, the proposed model was simplified to a model (differential equation) within the well-mixed population setting. This simplification allowed for an approximation and complete analytical analysis. However, in the well-mixed population model, consumers do not have constant reference groups. Instead, a reference group is sampled at random at each step, which is clearly at odds with the concept of a reference group, as these are relatively stable over time.

The primary goal of the current research is to extend the analytical well-mixed population model to an analysis of the model on common types of random networks. The primary method used for the analysis of the model’s behavior on networks is simulated. Technically, this paper offers two main contributions to the current research:The actual behavior of the model on all basic types of random graphs is quantitatively identical to the simplified differential model. Thus, the current study extends the applicability of the simplified differential model;The model’s behavior on networks with high connectivity and low clustering can be successfully predicted with the simplified differential model. Thus, the current study identifies the characteristic traits of networks that are especially difficult for modeling and predicting.

We can extend the understanding of the model’s behavior because the actual behavior of the model on random graphs is quantitatively identical to the behavior predicted by the simplified differential model. Thus, we can say that with the small reference group (on average), the result of the influence on market outcomes is small or not existent even for a strong influence. The market effects become stronger the larger the size of the reference group. Thus, for highly connected networks, such as some social networks, it seems possible to create a temporary prevalence of a single brand or opinion through the influence strength manipulation. This task is simpler in networks with a large size of the reference group.

The outline of the paper is as follows. In Section 2, the theoretical model is briefly described to introduce the notation and set the starting point. This section also contains the main results of the well-mixed population approximation presented in [35]. These results are further compared against the results of the simulations. Section 3 provides the detailed methodology used for the simulations and equilibria estimation. Section 4 discuses the results. Finally, Section 5 concludes.

## 2. Analytical Model and the Model on Graphs

A finite population N={1,2,…,N}, N∈N of consumers is assumed. Consumers are also referred to as either players, agents, or nodes. Each consumer can choose a single binary option. This option is thought of as a brand; see [35] for a thorough description. Thus, there are two brands, and each consumer may choose one of those. A state of a consumer n∈N is denoted by $c_{n}\in \left\{0,1\right\}$, where 0 denotes the first brand and 1 denotes the second brand. An *N* bit vector of consumer choices is called a profile of choices and is denoted by $c=(c_{1},\ldots ,c_{N})$.

There are $2^{N}$ different choice profiles; however, only a statistics *x* depending on a profile *c*, given as:(1)$$x=1-\frac{1}{N}\sum_{n}c_{n}.$$
is of interest. Statistics *x* is a share of consumers choosing the first brand. There are N+1 possible states of *x* as different profiles *c* yield the same states *x*. The set of all states *x* is finite and forms a grid {0,1/N,2/N,…,(N−1)/N,1}.

The above population model is considered in time t=0,δ,2δ,…, where δ>0 is a small time step. At each time step, a single consumer is chosen at random and makes a decision. A decision process (the decision process is described later, but the process considered depends only on a current state; hence, the behavior of the whole model is a Markov chain, cf. [35]) yields transition probabilities on profiles *c* and, consequently, transition probabilities on states *x*. Formally, let C denote the set of all profiles *c*. Because the size of the population is finite, the set of all profiles is also finite (of size $2^{N}$). Therefore, formally, we can take the set of all subsets of C as the σ-algebra. The probability of the transition from a profile *c* to a profile $c^{\prime }$ is denoted by $p_{c,c^{\prime }}$ and follows directly from the choice procedure described further. For any two profiles $c,c^{\prime }$, the transition probability $p_{c,c^{\prime }}=0$ if the Hamming distance between profiles *c* and $c^{\prime }$ is larger than one, because at each time step, only a single consumer may potentially change his/her choice. Consequently, given a state *x*, the new state $x^{\prime }$ after the transition satisfies $|x-x^{\prime }|\leq 1/N$, as at each step, only a single consumer may switch brands, and *x* is given by (Equation 1).

The decision process depends on a reference group of a given consumer. An undirected graph is used to model reference groups. An undirected graph G is a pair G=(N,E), where N is a finite set of nodes n∈N={1,2,…,N}, N∈N and E is a finite set of edges $e\in E=\{\left(n,n^{\prime }\right):n,n^{\prime }\in N\}$. Each consumer is thought of as a node in the graph G. Edges represent relationships between consumers. Given a consumer n∈N, his or her reference group is defined as a set of his or her direct neighbors:$$R\left(n\right)=\{n^{\prime }\in N:\left(n,n^{\prime }\right)\in E\}.$$

A selected consumer *n* decides on a brand to choose. The decision process is based on a reference group R(n) and an exogenous parameter α∈(0,1), called the follow-up probability. A consumer looks at a profile of choices of his or her reference group ${\left(c_{n^{\prime }}\right)}_{n^{\prime }\in R\left(n\right)}$ and determines which brand is in the majority. Afterward, he/she chooses the option in the majority with the follow-up probability α and the other option with the probability 1−α.

There are two elements to this process. The first one is checking the majority of his or her reference group. For small reference groups, it is a viable procedure. For larger reference groups, this part of the process may be questionable. In this case, the interpretation is that a given consumer uses his or her perception of the majority in his or her reference group.

The second element of the process is the follow-up probability. This exogenous parameter reflects the reference group’s influence strength on an individual. The neutral value of the follow-up probability is 1/2. If the value of the follow-up probability is above 1/2, this means that the reference group exhibits a positive influence on an individual. The negative effect is represented by the values of the follow-up probability below 1/2.

There may be quite different interpretations of the reference group and the follow-up probability. The reference group can be considered a sample from the general population (constant throughout a single simulation). A consumer is seeking anecdotal information based on a small sample from the general population with this interpretation. The other understanding is more in line with the meaning of the concept of a reference group as in social psychology. The reference group of an individual consists of his or her peers in a broad sense, e.g., school or work colleagues, business, trade, or art acquaintances, or individuals related in some capacity, with this interpretation. The follow-up probability may be interpreted as the measure of the attitude of an individual towards a reference group. The values above 1/2 are associated with a tendency towards homophily, while the values below 1/2 are considered a tendency towards heterophily. For simplification, it is assumed that all consumers share the same value of the follow-up probability.

The above procedure defines a Markov chain with states being all choice profiles *c*. The chain is irreducible because 0<α<1, and since for any choice profile *c*, the transition probability $p_{c,c}>0$, the chain is aperiodic, thus ergodic. This Markov chain translates into a Markov chain over the set of all states *x*. The Markov chain over the set of all states *x* is also ergodic.

In the above model on a graph, a reference group is constant in time, once the graph G of relationships is fixed. This model has been simplified in [35] by assuming that a reference group is not constant, but instead, it is randomly selected each time a decision is made and is always of size k∈N, where *k* is an exogenous parameter. Given a state *x*, associated with a choice profile *c* and a size *k*, this defines a binomial probability distribution over a number of consumers using the first brand in a reference group. Using this simplification, the probability of choosing a given brand by a consumer has been approximated, resulting in two probabilities: $p_{12}$ and $p_{21}$, where $p_{ij}$ is the probability that one consumer changes his or her choice from brand *i* to brand *j*. This in turn defines a Markov chain on states *x*, and this Markov chain is further approximated with a differential equation of the form, cf. [37],
(2)$$\dot{x}=p_{21}\left(x\right)-p_{12}\left(x\right).$$

Equation (Equation 2) for k=3 reads:(3)$$\dot{x}=\left(1-2x\right)\left(x^{2}\left(2\alpha -1\right)-x\left(2\alpha -1\right)+1-\alpha \right)$$
and for large *k* is approximated as:(4)$$\dot{x}=\left(\frac{1}{2}-x\right)+\left(\alpha -\frac{1}{2}\right)\mathrm{erf}\left(\frac{\sqrt{k}}{2\sqrt{2}}\frac{2x-1}{\sqrt{x(1-x)}}\right),$$
where erf is a standard error function (for a detailed discussion and derivation, see [35]). Both equations exhibit pitchfork bifurcation at a critical value of the follow-up probability $\alpha_{0}$ that depends on the size *k* of a reference group. For both Equations (Equation 3) and (Equation 4), there is an equilibrium at 1/2. For Equation (Equation 3), this equilibrium is asymptotically stable for α<5/6 and unstable for α>5/6. For α>5/6, there are two additional asymptotically stable symmetric equilibria. For Equation (Equation 4), the equilibrium at 1/2 is asymptotically stable for $\alpha <\alpha_{0}$ and unstable for $\alpha >\alpha_{0}$, where:$$\alpha_{0}=\frac{1}{4}\left(\frac{\sqrt{2\pi }}{\sqrt{k}}+2\right).$$

For $\alpha >\alpha_{0}$, there are two additional asymptotically stable symmetric equilibria. A formal proof was given in [35] and was based on [38] (p. 372).

## 3. Methods

Simulations were the primary tools in the current research. The model of a population, as described in Section 2, was simulated on random graphs. Since the model defines an ergodic Markov chain, it was possible to estimate stationary probability distributions for different types of random graphs and various values of parameters based on the simulations. Each such stationary distribution was further used to estimate equilibria that were compared to the equilibria achieved in the approximate differential model (for details on the connection between stationary distribution and equilibria of differential equations approximating the Markov chain, see [37]).

All simulations were performed on three types of random graphs. The first type used was the Erdos–Renyi random graph (ER) introduced in [39,40,41]. The second type used was the Barabasi–Albert random graph (BA) introduced in [42]. The last type of random graph used was the Watts–Strogatz random graph (WS) introduced and studied in [43,44,45].

A complete description of these types of graphs and their properties is beyond the scope of this paper. There are many great surveys, cf. [6,46,47,48]. However, there are good reasons to use these three types of random graphs. The first type of graphs considered in the simulations, the Erdos–Renyi random graphs (also called Poisson graphs), comprises graphs constructed by fixing a given number of vertices and then connecting each pair of vertices with the constant probability. We used Erdos–Renyi random graphs as the direct benchmark for the analytical model derived within the well-mixed population setting. Reference groups were selected at random in both the well-mixed population model and simulations. The difference was that in the well-mixed population, reference groups were selected at each step, while in the simulations, reference groups were selected at the beginning of the simulation and kept constant throughout. Nonetheless, the Erdos–Renyi random graphs seemed to be the closest in structure to the well-mixed population setting.

The Barabasi–Albert random graphs are the second type of graphs considered in the simulations. The structure of these graphs comes from the creation process. The process starts with a small graph, e.g., a cycle graph with three vertices. A single vertex is attached to the already existing graph with a fixed number of edges at each step. Those edges are connected to vertices with probability proportional to the vertices’ degrees. Thus, these random graphs are also called preferential attachment graphs. This is widely accepted as a model of citation networks, collaboration networks, the Internet, and other technological networks. For this reason, this type was selected for the simulations.

Finally, the Watts–Strogatz random graph was selected for the simulations. These graphs are constructed from circulant graphs with a fixed number of vertices and a list of jumps. In the second step of the process, each edge was rewired with a given value of the rewiring probability. This process led to a graph with a so-called small-world effect. Vertices far away from each other in the initial graph ended up relatively close to each other in the final graph due to the rewiring process. Thus, these graphs are also called the small-world graphs. The small-world effect is observed in many social networks, and this was the main reason for selecting this type of graph for simulations.

In both models, exact and approximate, there is the exogenous follow-up probability α, but the size *k* of a reference group is present only in the approximate model. To compare both models, the parameters of random graphs were chosen in such a way that the average degree of a node was close to the size *k* of a reference group. In particular, for each graph, two cases were considered. The first case was the small reference group case identified with k=3. The second case was the large reference group case identified with k=40. In each case, the size of the population was taken to be N=1000, and other parameters of the random graphs were set in such a way that the average degree of a node was close either to k=3 or k=40. In the case of Watts–Strogatz random graphs, the average node degree does not depend on the rewiring probability, but rather on the initial graph. For those graphs, four cases were considered: a small reference group and a large reference group with small and large rewiring probabilities, 0.1 and 0.9, respectively. Table 1 gives all parameter combinations used to generate random graphs.

For each combination of parameter values, given in Table 1, 1000 simulations, each of length $5\times 10^{4}$, were computed. For every simulation, a new graph was created from a given distribution. Figure 1a–c shows typical realizations of the simulation. In particular, Figure 1a,b shows typical realizations for low and high values of the follow-up probability α. Both figures suggest that the convergence to a stationary distribution is very fast, and the last 50% of the steps come from the stationary distribution. Figure 1c shows typical simulation realizations for values of the follow-up probability close to the critical value. In this case, transitions between the peaks of the stationary distribution are more frequent. Informed by these results, only the last 50% steps of each simulation realization were captured and used in the further analysis. That is, for each combination of parameters (including graph parameters and a value of the follow-up probability), a sample of size $25\times 10^{6}$ was created.

Based on the total sample for each combination of the parameters’ values, equilibria were estimated. For every combination of the parameters’ values, the first step was to estimate a probability density function of the stationary distribution. Then, a numerical procedure was used to approximate the peaks of the density function corresponding to asymptotically stable equilibria of differential Equation (Equation 2). Figure 1d shows the estimated density functions for all values of the follow-up probability. Approximating the peaks of a density function related to small or large values of the follow-up probability is simple. However, for moderate values of the follow-up probability, when density functions become relatively flat and with peaks close to each other, approximating the peaks presents a problem. For this reason, a simple bootstrap procedure was used.

First, note that the stationary distribution was symmetric around N/2 by the model’s design. The procedure estimated a probability density function *g* and then created a mixture probability density function $\tilde{g}\left(x\right)=1/2\cdot g\left(x\right)+1/2\cdot g\left(-(x-N/2)+N/2\right)$. The maximization procedure approximated peaks of the mixture probability density function $\tilde{g}$ in intervals [0,N/2] and [N/2,N] separately. The procedure was repeated 10 times with randomly selected subsets of simulation results to assess any potential inaccuracies. Thus, effectively, a simple nonparametric bootstrap technique was used.

The above procedure resulted in 10 peak approximations for any combination of the parameters’ values. These approximations allowed for a comparison with the theoretical values of asymptotically stable equilibria. Furthermore, the accuracy of the approximations can be assessed.

The whole simulation process and analysis of the results were performed with Wolfram Language Version 12.1. The entire code was divided into three parts. The first part implemented a single function modeling a single run of the simulation. The function took a graph with consumer choices as arguments. The function was compiled to an external library to boost the simulation speed. The second part of the solution used the compiled function to run simulations for various graphs with varying values of the exogenous parameters. This part of the code was not compiled as it had a minimal impact on the simulation times. The results of the simulations were stored as associations in native MX files. Finally, the last part of the code dealt with all the analysis, including equilibria estimations, figure generation, and other statistics. The whole process was run on a machine with Threadripper 2950x and 32 GB of RAM running under Linux Ubuntu 20.04.

## 4. Results and Discussion

As was mentioned before, for each graph distribution, two cases were considered: that of the small reference group and that of the large reference group. The only exception was the case of the Watts–Strogatz distribution. For this distribution, the additional parameter was the rewiring probability. The equilibria of the simplified analytical model were compared with the equilibria resulting from the simulations. Figure 2 shows bifurcation diagrams for all four cases. The follow-up probability is on the horizontal axes on all figures, while the vertical axes represent the phase space, that is the market share of the first brand. The orange color represents the case of the small reference group, and the blue color represents the case of the large reference group. Solid lines represent theoretical results. Orange dots represent the small reference group simulation results, and blue plus signs represent the results of simulations with the large reference group.

Erdos–Renyi random graphs are graphs randomly sampled from all graphs with the given number of nodes and edges. Therefore, they seem to be the closest to the simplified model. In the simplified model, each consumer uses a reference group of the same size *k* sampled each time a decision is made, while in the model on a graph, the reference group was randomly created at the beginning of the simulation and kept fixed. The size of a reference group follows a Poisson distribution. These two differences influence the behavior of the population and consequently the resulting equilibria.

Figure 2a shows the bifurcation diagram for the Erdos–Renyi graphs. The qualitative behavior of the simulations nicely corresponded to the analytical predictions. However, for the small reference group scenario, the theoretical model underestimated the critical value of the follow-up probability. For all values of the follow-up probability above the critical value, the theoretical model overestimated the position of the equilibria. For the large reference group scenario, the simulation results were identical to the theoretical model.

Barabasi–Albert graphs are built through the sequential attachment of new nodes where the edges are attached with probabilities proportional to the degree of already existing nodes. The algorithm leads to the distribution of the node degree being the power law (these graphs are also called scale-free graphs).

Figure 2b shows the bifurcation diagram for the Barabasi–Albert graphs. In the small reference group scenario, the predicted analytical behavior was not consistent with the observed behavior in two significant ways. First, the predicted value of the critical follow-up probability was too large. Second, the observed branches exhibited a linear behavior rather than the predicted nonlinear behavior. Consequently, the theoretical model underestimated the position of the equilibria near the critical value of the follow-up probability and overestimated the equilibria position for large values of the follow-up probability. In the large reference group scenario, the predicted behavior was a better approximation of the observed simulated behavior. Again, the predictions slightly overestimated the critical follow-up probability.

The last type of random graph considered is the Watts–Strogatz random graph. These graphs are constructed through a process of edge rewiring. The construction algorithm starts with a circulant graph with *N* vertices and a jump *j*. Then, each edge is rewired with a given probability. The average node degree does not depend on the rewiring probability and is defined by the jump *j* of the initial circulant graph. For this reason, four cases were considered: all combination of the sizes of the reference group (small and large) and rewiring probabilities (0.1 and 0.9).

Figure 2c,d shows the bifurcation diagrams for the low and high rewiring probability, respectively. In the small reference group scenario and a low rewiring probability (Figure 2c), the predictions of the analytical model were different from the results achieved in the simulations for almost any value of the follow-up probability above 5/6, that is the theoretical, critical value of the follow-up probability. The analytical model profoundly underestimated the critical value, leading to overestimating equilibria positions for high values of the follow-up probability. If the rewiring probability was increased (Figure 2d), the simulation results were almost in line with the theoretical predictions. The theoretical model only slightly overestimated the critical value of the follow-up probability. In the large reference group scenario, the theoretical model offered quite precise predictions. This was certainly true for the high rewiring probability. For the low rewiring probability, the theoretical model underestimated the critical follow-up probability, which led to an overestimation of the equilibrium positions for the moderate values of the follow-up probability.

The main observation from the results described above was that the theoretical model was correct as far as the qualitative behavior was concerned. In all cases, we can see pitchfork bifurcation. Furthermore, for small reference group cases, the critical value of the follow-up probability was lower than the critical value of the follow-up probability for large reference group cases.

The discrepancies start to show when we go from the qualitative behavior to the quantitative behavior. Comparing the results of the simulations for all types of random graphs, we can see that the theoretical model was most of the time correct as far as the large reference group model was concerned. The only exception was the Watts–Strogatz graph with a low value of the rewiring probability. In this case, the behavior of the theoretical model near the critical value of the follow-up probability was not accurate.

This was different for the small reference group case. Here, for all types of graphs, we can observe some inaccuracies. However, the worst situation was in the case of the Watts–Strogatz graph with a low rewiring probability.

We start by looking at the mean clustering coefficient. Figure 3 shows a box and whiskers chart of the mean clustering coefficient for all cases. Two observations can be made. First, the mean clustering coefficient is larger by an order of magnitude for the Watts–Strogatz graph with a low rewiring probability, thus suggesting that the poor predictions of the theoretical model are connected to a high level of clustering. This does not tell the whole story, as the mean clustering coefficient is (slightly) higher for the large reference group case for each graph. However, the predictions for large reference groups were, in general, better.

The second statistic of interest is betweenness centrality. Figure 4 shows the histograms of betweenness centrality for all four cases. The small reference group cases are marked with the orange color, and the large reference group cases are marked with the blue color. The first observation is that there were far more nodes with high betweenness centrality in the small reference group cases. Thus, this suggests that the inaccurate predictions of the theoretical model had to do with many central nodes.

In the end, it seems that the problematic networks are those that have high clustering and many central nodes. However, the precise understanding of the influence of a graph structure on the position of equilibria is a very complex issue and is well beyond the scope of this paper.

## 5. Conclusions

Several conclusions can be drawn from the above comparison. Firstly, the simulation results exhibited qualitatively the same behavior as the proposed theoretical model. For each type of random graph distribution, there were two distinct phases: one was unimodal for α lower than the critical value, and the other was bimodal if α had values above the critical value. In the theoretical model, there was a pitchfork bifurcation at the critical value of the follow-up probability.

The precision of the predictions of the analytical model was different in different situations. It seems that for the large reference group, the predictions were pretty good in both phases. The model overestimated (Barabasi–Albert graphs) or underestimated (Watts–Strogatz graphs with the small rewiring probability) the critical follow-up probability slightly. However, in the other cases, the fit was almost perfect.

The predictions for the small reference group were of lower quality. The critical value of the follow-up probability was either overestimated, as was the case for the Barabasi–Albert graphs and Watts–Strogatz graphs with a high probability of rewiring, or underestimated, as was the case for Erdos–Renyi and Watts–Strogatz random graphs with a small probability of rewiring.

The general conclusion from the above research is that the results valid for the simplified model extend beyond the well-mixed population setting. The qualitative results were identical. There were still two distinct phases. If the tendency towards homophily was low, there was only one stable symmetric equilibrium. Once the tendency towards homophily was strong, a single stable equilibrium morphed into two symmetric stable equilibria. Thus, the market share of the first brand oscillated around one of the stable equilibria. In other words, if we look at the isolated effect of a reference group, it may push the market equilibrium towards one of the equilibria, giving one of the brands an advantage. This effect is reversible (as long as we consider the isolated impact), however long it takes.

The move from one phase to the other happened at a critical value of the follow-up probability. The critical value was relatively high for the small reference group and relatively low (close to 1/2) for the large reference group. Thus, for the third party manipulating values of the follow-up probability, it was easier to push the system into the bistable phase if the size of the reference group was large. This result perhaps explains why social networks try to incentivize users to have many so-called “friends” as it is easier to manipulate consumers’ choices in such cases. This result is in-line with the theoretical finding and analysis of real networks [49].

The current research had many limitations that can be divided into two main categories. The first type of possible further research would consider theoretical modeling to deliver a purely analytical model connecting a network’s topology precisely with the system’s dynamics. This, however, seems to be a challenging task. The second type of further research would consider a more elaborate simulation framework. For example, in the current study, all consumers in the population shared the same value of the follow-up probability. It is technically possible to extend the simulation to cover the distribution of the follow-up probability. Such a simulation would cover scenarios with a population having different responses towards the reference group. Another possible extension is to explore more general models of graphs’ topologies. All these, and perhaps other, extensions are left for future work. 

## Figures and Tables

**Figure 1 entropy-23-01007-f001:**
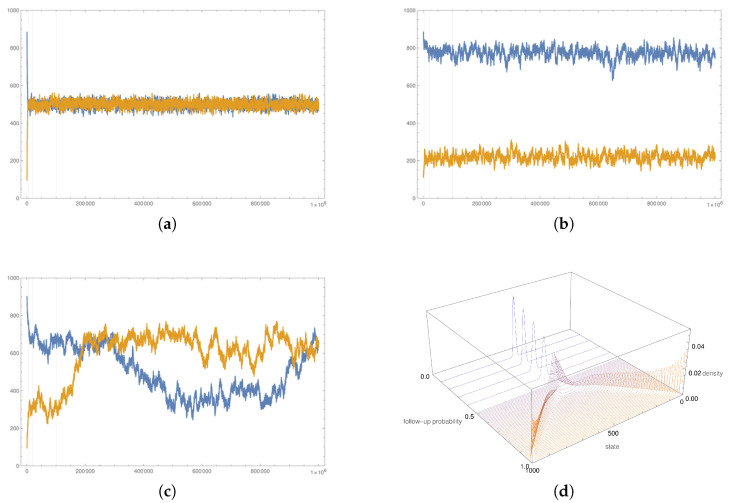
Typical realizations for various follow-up probability values. Figures a–c show two separate simulations’ realizations with different initial shares. Blue lines are realizations with a prevalent initial share of the first brand, and orange lines are realizations with a dominant initial share of the second brand. The horizontal axes show the time steps, and the vertical axes show the number of consumers choosing the first brand. Figure d shows stationary distributions for various values of the follow-up probability. The first three figures show the typical behavior of the simulation for values of the follow-up probability below the critical value (**a**), above the critical value (**b**), and close to the critical value (**c**). (**d**) shows how these modes of behavior are reflected in the stationary distributions.

**Figure 2 entropy-23-01007-f002:**
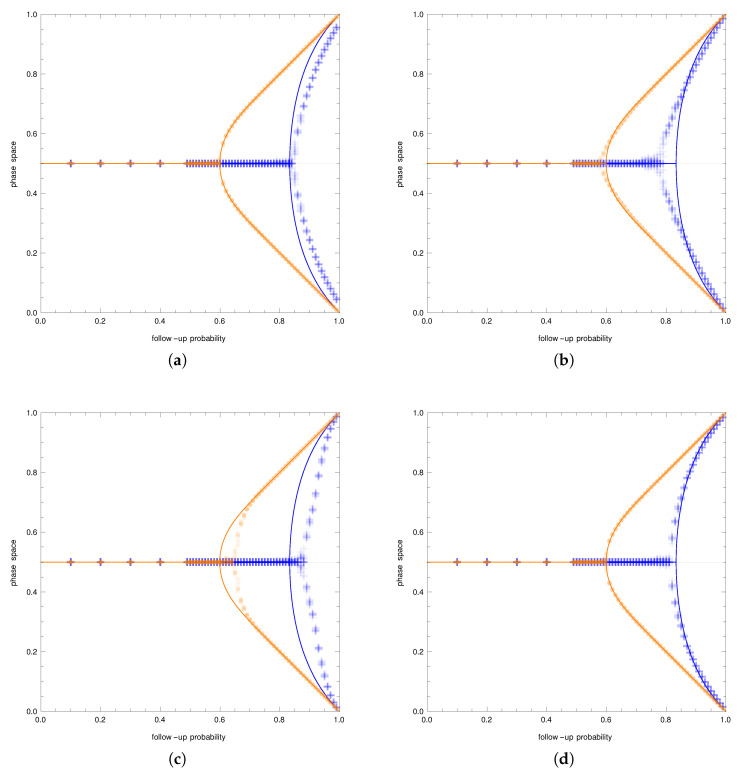
Results of the simulations. The solid lines depict the analytical results, while points and crosses show the simulation results. The blue color is for the small reference group, and the orange color is for the large reference group. The horizontal axes show the values of the follow-up probability, while the vertical axes show the market shares of the first brand. The figures show bifurcation diagrams with the simulation results on top for Erdos–Renyi graphs (**a**), Barabasi–Albert graphs (**b**), and Watts–Strogatz graphs with the small value of the rewiring probability (**c**) and the large value of the rewiring probability (**d**). Qualitatively, the analytical and simulation results coincide.

**Figure 3 entropy-23-01007-f003:**
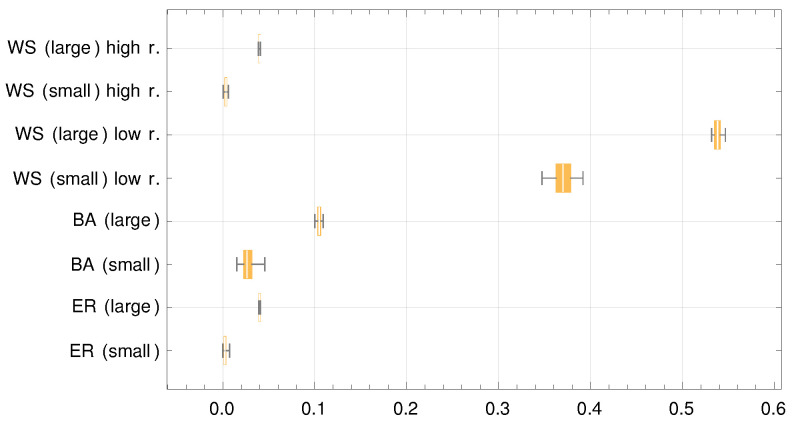
Mean clustering coefficient. The horizontal axis shows the values of the mean clustering coefficient, while the vertical axis shows different cases. The words in parentheses, small or large, reflect the reference group size. For Watts–Strogatz graphs, low r. and high r. mean a low and high value of the rewiring probability, respectively. Values for the Watts–Strogatz graphs with the small value of the rewiring probability are larger by an order of magnitude.

**Figure 4 entropy-23-01007-f004:**
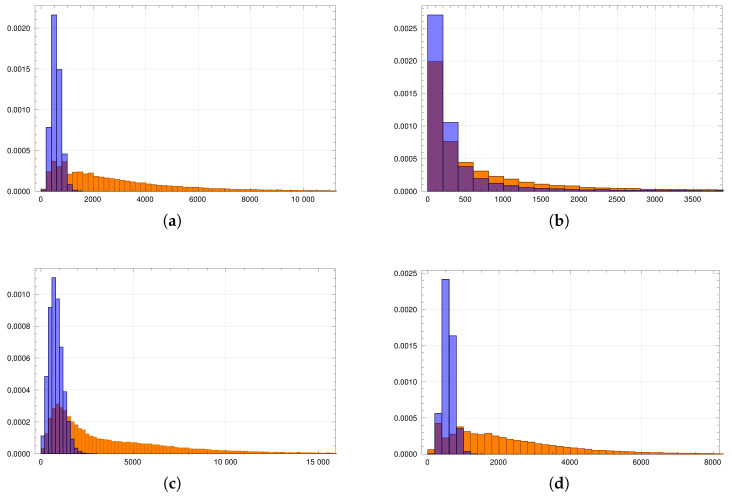
Betweenness centrality distributions. The horizontal axes give betweenness centrality, while the vertical axes give the frequencies. The orange color depicts graphs with the small reference group, and the blue color represents the graphs with the large reference group. The figures show the results for the Erdos–Renyi graphs (**a**), Barabasi–Albert graphs (**b**), and Watts–Strogatz graphs with a small value of the rewiring probability (**c**) and a large value of the rewiring probability (**d**).

**Table 1 entropy-23-01007-t001:** Random graphs’ parameters. The labels are ER for Erdos–Renyi, BA for Barabasi–Albert, and WS for Watts–Strogatz.

Graph Type	Parameters	Mean Node Degree
ER	number of edges = 1500	3.0
ER	number of edges = 20,000	40.0
BA	degree of attached node = 2	4.0
BA	degree of attached node = 20	39.6
WS	rewiring probability = 0.1	4.0
	initial graph size = 2	
WS	rewiring probability = 0.9	4.0
	initial graph size = 2	
WS	rewiring probability = 0.1	40.0
	initial graph size = 20	
WS	rewiring probability = 0.9	40.0
	initial graph size = 20	

## Data Availability

The data presented in this study are available on request from the corresponding author. The data are not publicly available due to the size.

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
