# Peer review of "Market Choices Driven by Reference Groups: A Comparison of Analytical and Simulation Results on Random Networks"

_entropy, 2021, doi:10.3390/e23081007_

Round 1
Reviewer 1 Report
I'm made comments for the author in the attached pdf The paper was written by an insider for an audience of insiders. It needs to add more explanation about why this article is important to the average reader and more explanation on the results, especially a better explanation of the Figures.
Author Response
Please, find my response in the attached PDF file.

Reviewer 2 Report
After carefully reading the work by M. Ramsza, entitled: "Market Choices Driven by Reference Groups. Comparison of Analytical and Simulation Results on Random Networks," I regret to inform that I cannot recommend it for publication in Entropy.
The main reason is that, in its current form, the manuscript presents itselfs as a simple computational excercise of an analytical model. In fact, the author refers to his own manuscript as an "excercise." Furthermore, the introduction is so short that it is impossible to place the work in context:
1) Why is the model presented in section 2 important for the community?
2) Why the author decided to explore the Erdos-Renyi, Barabasi-Albert, and Watts-Strogatz graph models? There is not clear explanation why should we care about them. The author only refers the reader to Refs. [27-29]. I believe a more detailed description of these models, and why they might be important for the community, is needed.
3) Finally, I would like to see an explanation why the author decided to work on a computational verification of an analytical model. The process is typically the other way around.
Author Response

(The authors gave the same response as above.)

Reviewer 3 Report
This is an interesting paper with theory and experiments balanced. The paper focuses on market choices driven by reference groups with a valid motivation. The presentation of the paper can be improved. Some comments are as follows.
1. The decision process of each consumer is based on his reference group given as a group of his direct neighbors. Do all neighbors have the same weight in decision making?
2. The main contribution should be summarized in the introduction. I suggesting adding a comparison table to show the difference of the current paper and the existing works.
3. The paragraph between line 52 and 58 is not clearly written. What is the probability space you are considering? Please give a formal definition with measure and sigma algebra.
4. Both equations exhibit pitchfork bifurcation at a critical value of the follow-up probability a that depends on the size k of a reference group. This statement needs to be justified. It is not obvious at all in the current model.
5. The mixture used in line 122 is enigmatic. It does not seem to be right as the scaling go out of the boundary. Please double check.
6. In line 247-249: in fact this is due to clustering effect in networks. It is shown in the paper 'Clustering coefficients of large networks' the typical clustering of the networks you considered in the current paper. This actually explains the observation.
Author Response

(The authors gave the same response as above.)

Reviewer 4 Report
The manuscript covers an interesting R&D topic and fits the scope of the Journal. Nonetheless, the paper requires extra efforts to improve its quality and presentation. A set of comments are expounded hereafter.
- Regarding the format of the document:
Table 1 includes a set of abbreviated names of the graph types, such as ER, BA, and WS; however, within the text there is no mention to the correspondence of the abbreviations and the entire denominations of the graphs. It is easy to suppose that ER stands for Erdos-Renyi, but the reader is not expected to suppose this type of information.
The title of the fifth section must be revised, namely, “Colusions” should be replaced by “Conclusions”.
Sections like Authors contributions, Funding, etc., are missing.
The format of references must be slightly revised following the template of the Journal. Namely, the abbreviated names of journals must be used.
- About the content of the manuscript, as aforementioned, it covers an interesting topic. The comments after a careful revision are the following:
The keyword “games on graphs” is not found in the manuscript; therefore, its suitability as keyword should be revised.
There are not references of recent years, the most modern corresponds to 2015. It is evident that the contextualization must be enhanced by reviewing state-of-the-art publications dealing with the topic of the manuscript in order to highlight its relevance and opportunity. For example, some recent works, published in MDPI, which emphasize the applicability of random graphs are now given for consideration by the author:
- Muñoz, H.; Vicente, E.; González, I.; Mateos, A.; Jiménez-Martín, A. ConvGraph: Community Detection of Homogeneous Relationships in Weighted Graphs. Mathematics 2021, 9, 367. https://doi.org/10.3390/math9040367
- Kłopotek, R.A. Modeling Bimodal Social Networks Subject to the Recommendation with the Cold Start User-Item Model. Computers 2020, 9, 11. https://doi.org/10.3390/computers9010011
- Baxter, G.J.; da Costa, R.A.; Dorogovtsev, S.N.; Mendes, J.F.F. Filtering Statistics on Networks. Entropy 2020, 22, 1149. https://doi.org/10.3390/e22101149
The third section should include at least a brief explanation about the software environment and/or language used to implement the expounded graphs. Details about versions as well as additional modules/toolboxes should be also considered. This information could be useful for the interested reader and enhances the description of the methodology.
The caption of the fourth figure should include a brief description of what is contained in each panel, as it is indicated in the template of the Journal.
The expression “It is a different story for …” (line 202) is not adequate, from this humble reviewer viewpoint, for a high-quality scientific paper.
The Results and discussion section should include certain discussion about the comparison of the results with previous (recent) literature in order to prove that the paper contributes to the body of knowledge. The novelties and strengths of the manuscript should be clearly highlighted in such a section as well as in the Abstract and in the Conclusions in order to attract the attention of the reader.
Moreover, the Conclusions section could be enriched by adding some future guidelines that the author is considering on the view of the achieved results. For example, the implications in social networks could constitute a further research work.
Author Response

(The authors gave the same response as above.)

Round 2
Reviewer 1 Report
Figure 4, 1st line, add 1 space between the end of a sentence and start of new one: "distributions. The"
line 357: delete "either" or revise the sentence
line 369: revise to "findings and analyses"
Reviewer 2 Report
The author has addressed all my concerns. Specifically, the author has now put a lot of effort placing the work into context, explaining the reasons for selecting the studied graph models. I am happy to recommend the revised version of the manuscript for publication in Entropy.
Reviewer 3 Report
I have no further comment for this version. It can be accepted.
Reviewer 4 Report
The provided suggestions have been properly addressed.